

# Water stable isotopes (δ²H and δ¹⁸O) in the Peninsula of Yucatan, Mexico.

Eduardo Cejudo[1], Gilberto Acosta-Gonzalez[1], Rosa Maria Leal-Bautista[1] and Hector Estrada-Medina[2].

[1]CONACYT - Centro de Investigación Científica de Yucatán A.C., Water Sciences Unit. Calle 8, No. 39, Mz 29, SM 64. Cancun, Quintana Roo, 77524. Mexico.
[2]Universidad Autónoma de Yucatán, Departamento de Manejo y Conservación de Recursos Naturales Tropicales, CCBA. Km 15.5 Carr. Mérida-Xmatkuil. Mérida, Yucatán, A.P. 4-116. Mexico.

*Correspondence to*: Eduardo Cejudo (eduardo.cejudo@cicy.mx)

**Abstract.** The hydrogen and oxygen isotopic composition of water is a very important tool to estimate water balance, groundwater recharge, and evaporation. Water isotopes have been used to increase our understanding of the distribution and amounts of renewable and non-renewable groundwater. Isotopic data from precipitation and groundwater is available in much of Mexico but there is little information from the Peninsula of Yucatan, an area heavily relying in groundwater in which current estimates of groundwater availability are uncertain. In this paper, we compiled published and unpublished δ²H and δ¹⁸O data in meteoric (waters derived from precipitation), ground- and pore-waters, to obtain a regional meteoric water line (RMWL) expressed by the equation $\delta^2 H = 8.1846\ \delta^{18} O + 10.289$. The data suggest that precipitation originates in convective systems, low-pressure events, moisture from frontal events, and from re-condensed moisture. The evaporation lines from groundwater suggest mixing of water with different isotopic composition, but also provide clues to recent meteoric water rapid recharge, likely from rain events of great intensity. We present a groundwater isoscape of the Peninsula of Yucatan and finally address the lack of conciliation between hydrogeology and groundwater management.

## 1 Introduction

Isotopes are nuclides that have the same atomic number but different number of neutrons and mass (IUPAC 1997). The isotopic composition of a compound is defined by that variation in the number of neutrons of each element, which imparts a slightly different atomic weight (Kendall and Caldwell, 1998). The environmental (or stable) isotopic ratios of water, $\delta^{18}O$ and $\delta^2H$ (or δD, standard notation for deuterium) have been used to identify the origin of groundwater, aquifer recharge, geochemical reactions and soil-water-plant-atmosphere interactions (Clark and Fritz, 1997; Gat, 2010). The isotopic composition of water is useful for estimating accurate water balances (Gat 2000). For instance, one of the most used tools for estimating groundwater recharge is the isotopic composition of meteoric water, the water derived from precipitation. The most common representation of the isotopic composition of meteoric water is a δ-plot, a 2-dimension scatter plot ($\delta^{18}O$ versus $\delta^2H$) that reveal the Global Meteoric Water Line (GMWL, Craig, 1961); in which the dispersion of oxygen and hydrogen isotopic composition has a slope





of 8, and the intercept has been named the d coefficient (≈10 Dansgaard, 1964). The d coefficient represents the conditions at which the vapour was formed; deviations from this line are controlled by evaporation or condensation at different temperatures (Mook, 2002), non-equilibrium evaporation (Clark and Aravena, 2005), or re-evaporated precipitation (Peng et al., 2010). At smaller scales, a Regional Meteoric Water Line (RMWL) or Local Meteoric Water Line (LMWL) is particularly useful to

define the range of input parameters, and to estimate infiltration and recharge. Isotopes are essential for building an improved understanding of the spatial distribution and amounts of non-renewable or fossil groundwater (Kendall and Doctor, 2003). A significant amount of isotope data from aquifers worldwide exists today, with most of the data collected as part of International Atomic energy Agency IAEA technical cooperation projects, that have been integrated into continental-scale evaluations of water resources (Aggarwal et al., 2005).

### 1.1 The use of stable isotopes of water in the Peninsula of Yucatan


Mexico has had important studies of precipitation and groundwater isotopes in the central area of the country (Cortes and Farvolden, 1989; Cortés et al., 1997; Edmunds et al., 2002; Peñuela-Arévalo and Carrillo-Rivera, 2013). However, there is little information regarding the isotopic composition of water in the Yucatan Peninsula (YP). Most of the isotopic data has been obtained from sinkholes, locally known as cenotes (Socki et al., 2002; Wassenaar et al., 2009; Haukebo, 2014; Lases-

Fernandez et al., 2019), lakes (Curtis et al., 1996; Perez et al., 2011; Hodell et al., 2012), and coastal lagoons. Some studies have used isotopic data for estimating groundwater recharge and residence time (Marín et al., 1990; Perry et al., 2003). Waasennar et al. (2009) generated groundwater isoscapes nationwide. An isoscape (or isotopic landscape) is representation of the spatial-temporal distribution of isotopes in any environmental compartment (Bowen, 2010). Recently, the isotopic composition of rainwater and a local meteoric water line has been obtained for the northern area of Quintana Roo (Lases

Fernández et al., 2019). However, heavier sampling is required over the rest of the Yucatan Peninsula to improve our current water estimates in a region where groundwater is the primary source and whose water balance has not been verified nor calibrated.

In this paper, we compile available published information, together with unpublished data of the stable isotopes of water ($^2$H

and $^{18}$O) in five hydrosphere reservoirs (meteoric, groundwater, lake, coastal lagoons and pore water). We propose a RMWL for the northern YP and evaporation lines for groundwater. Finally, we address the need of reconcile the hydrogeology with the groundwater management and how stable isotopes can provide valuable information.

### 2 Methods

### 2.1 Isotopic data in the Peninsula of Yucatan

We data mined published papers, dissertations, proceedings, databases, and websites together with unpublished data of oxygen and hydrogen isotopes of water ($\delta^2$H and $\delta^{18}$O) in the three states of Mexico comprising the YP: Campeche, Quintana Roo and



Yucatan (Fig. 1). Datamining was conducted using search engines Google Scholar, Scopus, Internet Archive, Web of Science, and the "Virtual library and Catalogue" at www.cicy.mx/biblioteca/biblioteca-virtual; using the search terms *Isotopes*, *Water, Yucatan, Campeche* and *Quintana Roo.* We focused on water isotopes only and did not including data from sediments, shells, or other matrices. All data was organized following the template of the Global Network for Isotopes in Precipitation GNIP. We included all available information: location (state), geographic coordinates (decimal degree), author or authors, type of water reported (meteoric, groundwater, sea, lake), date (month-year) $\delta^{18}$O, $\delta^{2}$H, *d*-excess, altitude (m above sea level), aquifer, historical annual precipitation (in mm), soil type, hydrogeological sub-region, geology (type and class) and analyses providers (*Supplementary Material 1*).

Isotopic data from meteoric waters was used to create a RMWL with data from locations in northern YP (north of the latitude 20° N). Data from groundwater, seawater, coastal lagoons, and lakes was included in the RMWL for comparison purposes. We examine evaporation lines by plotting the groundwater isotope composition considering the states geopolitical division. In cases where only $\delta^{18}$O was reported, we used the equations obtained from evaporation lines for each state in order to interpolate missing $\delta^{2}$H data (*Supplementary Material 2*).

## 2.2 Geographic Information System.

Vector data (*.shp* format) and table-form information for historical precipitation (from years 1910 to 2009), hydrological regions, basins and sub-basins was obtained from INEGI (National Institute of Statistics and Geography, Mexico) and CONABIO (National Commission for the Knowledge and Use of Biodiversity, Mexico). The federal government grants the rights for groundwater abstraction as concessions. Concession volumes and geographical location were retrieved from the website http://datos.aguaparatodos.org.mx/concesiones/. All data was processed, and maps were produced using the free software Quantum GIS (QGIS 3.8). We used historical annual precipitation (1910-2009) to create precipitation contours from 600 to 1800 mm and then we superimposed the number of concessions and administrative boundaries. For the isoscape map, contour lines of $\delta^{18}$O and $\delta^{2}$H were obtained by interpolation of data collected for this study. The Weighted Inverse Distance (IDW) method was applied to the database to obtain an interpolation and subsequently the contour lines were generated using the triangular interpolation method (TIN).

## 3 Results and Discussion

The data in this paper comprises meteoric water (n=130), groundwater (n=213), lakes and coastal lagoons (n=128). One of the main observations is that the geographic distribution of the data is not spatially homogeneous; 51% of the data correspond to the states of Quintana Roo and 46% to Yucatan; the state of Campeche has had little sampling effort so far (Fig. 1). The focus of research on water isotopes has been mainly on groundwater (44%), with meteoric water (27%) and lakes (27%) studied less. The uneven distribution of sampling effort highlights two particular situations. First, the isotopic data compiled in this paper





were collected for a variety of scientific studies whose effort has been dedicated to geochemical and paleoclimate studies, which use sediments but lake water is collected at the same time. Second, groundwater extraction is different among states; thus, sampling efforts have been focused on that matrix. The state of Yucatan accounts for 46% of the extracted volume; whereas Campeche and Quintana Roo extracts around 28% and 26%, respectively. Additionally, Campeche relies also on surface water for human consumption; thus, groundwater sampling has not been as intensive as other regions.

### 3.1 Regional Meteoric Water Line of the northern Peninsula of Yucatan

The available meteoric water isotopic data (n=130) allowed us to create a RMWL of the northern area of the YP (Fig. 2), Eq. 1 (r$^2$=0.9413):

$$\delta^2 H = 8.1846 \, \delta^{18}O + 10.289 \tag{1}$$

Our RMWL compares very well with the local meteoric water line from northeast coast of Quintana Roo (20°35.2', -87°8.04') determined by Lases-Fernandez et al. (2019) ($\delta^2$H=8.17 $\delta$ $^{18}$O+11.698) with no difference in the slopes (t=-0.34, p=0.37). Although both agree well, we propose to use Eq.1 as the RMWL of the Northern Peninsula of Yucatán given that it includes sampling points from other longitudes. We stress the regional demarcation because most of the data compiled in this paper represents locations northern than latitude 20 °N. There are very few data points representing the central-south area of the territory (see Fig. 1).

The $\delta^{18}$O distribution of the data (from 0.83 to -9.7 ‰) likely represents seasonal variability and meteorological conditions in which the precipitation formed. Lower $\delta^{18}$O values (and lower $\delta^2$H values) correspond to larger rainouts of the air mass, observed as depletion of the heavier isotope ($^{18}$O) (and $^2$H) in rain in both seasons, summer and winter (Rozanski et al. 1993). Tropical storms also deliver rain that is $^{18}$O depleted in comparison to regular precipitation (Lawrence and Gedzelman 1996; Perry et al 2003). Data with *d*-excess close to the global average (10 ‰, Fig. 3) is assumed to represent evaporation from the average source (*i.e.* oceanic water) at 25° C of seawater temperature and relative humidity of 80% (Daansgard, 1964; Merlivat and Jouzel, 1979). *d*-excess lower than 10‰ could be from convective systems or low pressure precipitation events (Guan et al., 2013); whereas *d*-excess higher than 10‰ might represent precipitation from air that has undergone more than one condensation, moisture originating from frontal events (Clark and Fritz, 1997; Guan et al.; 2013), or convective recycled moisture (Wassenaar et al., 2009). In the YP convective rains occur more during the dry season, resulting in depleted $^{18}$O and $^2$H; cold front events common from November to March would result in *d*-excess higher than 10‰. The wide distribution of *d*-excess in wet season (July to October) and the increase in isotopes deltas at different times of the year can be attributed to the entrance of different air masses with variable moisture (Li et al.; 2015).

According to Daansgard (1964), the difference between $\delta^2$H in summer months and winter months ($\delta$s-$\delta$w = -3.56) define the north of the YP as "continental station at low latitude". Under some circumstances, the evaporation from falling droplets would modify the trend of the line, which then might resemble evaporation lines of surface waters (Gat, 2010). Occasionally, meteoric water lines join separate clusters of data or rain events associated with air masses of different origin (IAEA, 1992). Under those circumstances, the best-fit line for the data might not be completely accurate. With the information available, we assume





that this RMWL truly represents the land surface, for the data represent the northern portion of a region with similar climatic conditions and it is similar in slope to the GMWL. However, its validity is still spatially limited (Gat, 2005). We do not fully explore the amount effect because the precipitation *in situ* was not systematically recorded, or it was not measured close to the sampling point. When exploring the altitude effect in cross sections, we observed a slight trend in a north-south axis due to the
low relief (Fig. 4). Perhaps sampling the longest N-S cross-section, including the highest altitudes in the peninsula (≈ 120 m in the Ticul range and ≈220 m in the south), might yield a trend as the one observed in Northern Central America (Lachniet and Patterson 2009).

## 3.2 Groundwater

Groundwater has been slightly more intensively sampled; however, few studies analyzed its isotopic composition and most
reports represent single sample analysed for specific purposes. Few studies have been designed for systematic groundwater measurement. A study in the area around San José Tzal, Yucatan (20.83 °N, -89.650 °W), revealed that the groundwater isotope composition is similar to the meteoric water in the first half of the year when sampling was done in both reservoirs (unpublished data, Fig. 5). This trend suggest that groundwater retains the meteoric isotopic signature of that precipitation, which suggest fast recharge after precipitation. Unfortunately, we do not have more information that allow us to speculate preferential summer
or winter recharge with the data available.

A large number of the groundwater samples collected in the three states of YP does not follow the RWML, partly because it represents the north of the YP, but data also suggest evaporative effects (Fig. 6). The groundwater in the three states of YP follows different evaporation lines with slopes and *d*-excess different from the existing RMWL. This suggest that infiltration could have occurred with evaporative loss (Clark and Fritz 1997; Wassenaar et al 2009). Unfortunately, there is not enough
information for achieving better resolution or deeper speculation. Considering that groundwater is a long-term integrator of precipitation and infiltration, the shift of the groundwater samples down and to the right of the RMWL indicate evaporation and mixing of water with different isotopic composition. After precipitation, the passage of meteoric water through the surface and vadose zone might result in partial evaporation from open surfaces or in the soil column (Gat, 2010).

We assume that water moving into the aquifer after runoff, evapotranspiration, and soil retention represents the local recharge
flux. The water that infiltrates into soil will mix with previously existing moisture in the vadose zone, changing the original isotopic composition of the rainwater. Thus, it can be assumed that intense rain events are more represented in recent recharge flux (Gat 2010, Lases-Fernandez et al., 2019). Once reaching the aquifer, the recently infiltrated water with an isotopic composition slightly different form the RWML mix with the existing groundwater. It has been suggested that, in thin vadose zone (<10 m), meteoric water moves fast but with differential contribution to the isotopic composition of the local aquifer
(Lases-Hernandez et al., 2019). However, it is not clear if water moves faster or only that there is less distance to travel relative to the local phreatic level. We argue that deviations of the isotopic composition of groundwater from the RMWL is the result of mixture of meteoric water with water of different isotopic composition. Following intense precipitation, the actual freshwater lens increases rapidly due to thin soil layer, almost non-existent runoff and rock with high permeability. This





situation entails that residence time in the vadose zone is short and that shallow aquifers would retain the isotopic composition
greatly similar to meteoric water. To the best of our knowledge, residence time in the shallow aquifer has been estimated up
to three years (Lases-Fernandez et al., 2019) with water and dissolved substances traveling horizontally 30 km in about four
years (Martinez-Salvador et al., 2019). Such variability is likely associated with the characteristic of the epikarst (Aquilina et
al., 2006; Trček, 2007), high permeability (Steinich and Marin, 1997), preferential flow (Martinez-Salvador et al., 2019), and
possibly point recharge (Somarante, 2014), among others. Perry et al. (2003) found that after a couple of years, the aquifer in
the northwestern area of the YP was still recovering to its groundwater isotopic baseline after being hit by Hurricane Mitch
(1998, $\delta^{18}O$=-9‰, $\delta^{2}H$=-60‰). Tropical storms and hurricanes will impart a very depleted water isotopic composition that
can be traced and used to estimate recharge. The limited, non-systematically collected evidence suggests that the stable isotope
composition of groundwater in some locations of the YP might change in response to intense rain such as tropical storms and
hurricanes, and its effect last some time. Greater sampling intensity is required at a larger scale, particularly sampling of
meteoric water, in order to know the variability in the isotopic composition as it is the mayor source of water into the aquifer.

**3.2 Groundwater isoscape**

Previous efforts (Waasennar et al., 2009) represented the groundwater isoscape of the YP with low spatial and temporal
resolution. The isoscape presented here shows more depleted values in the areas with lakes and sinkholes exposed to the
atmosphere, likely the result of intense evaporation (Fig. 7). Large portions of the data corresponding to the ring of sinkholes
(a zone in the northern PY, more intensively studied that the southern region) have $\delta^{18}O$ values between 0 ‰ and -2‰. We
consider that sinkholes are the expression of the local phreatic level; yet, they might actually have a water isotopic composition
more similar to the meteoric water because of fast recharge or water mixture due to surface interception. It is necessary to
increase the amount of data from the southern portion of the peninsula given the need of accurate estimates of water availability,
differences among regional precipitation and the complex patterns in which water infiltrate and flow in different areas of the
peninsula (Bauer-Gottwein et al., 2011; Sandoval Montes and Heredia Escobedo, 2018)

**3.3 Tracing sources of water**

Water isotopes have been used to identify the sources of water by specific plant species assuming that there is not isotopic
fractionation during plant water uptake (Dawson et al., 2002). With the $\delta^{18}O$ of groundwater and tree stem water (sap), it has
been found that perennial and deciduous trees in northern Yucatan (Quejereta et al., 2006) and Quintana Roo (Hasselquist et
al., 2010), overcome seasonal water limitations by alternatively using water from the aquifer, water stored in the vadose zone
(the upper 2.5 m of the soil profile), and even water from unconsolidated rock (locally named sascab). Sascab, a soft rock, can
hold up to 3% of gravimetric water during the dry season (Estrada-Medina et al., 2010). The same approach to identifying
water sources was used by Quejereta et al., (2007) to investigate water used by mature tree individuals of deciduous and





perennial species. They found that all trees relayed on water from the upper 2-3 meters of the vadose zone in both dry and wet season, but perennial species can probably penetrate deeper through pores, fractures, and cavities in the rock, reaching deeper sources of water. Estrada-Medina et al., (2013) also traced the sources of water utilized by two deciduous species (*Gimnopodyum floribundum* and *Piscidia psipula*); they found that both species relied on soil as their main source of water

during the wet seasons. During the dry season *P. piscipula* relies more on soil water while *G. floribundum* relies more on water from bedrock. Interestingly, during the precipitation events during cold fronts, *P. piscipula* had $\delta^{18}O$ different from that of the soil and bedrock, suggesting that this species is able to take advantage of condensate water, at least during that season.

**3.5 Using isotopes for reconcile hydrogeology with groundwater management**

We do not have dependable information regarding the amount of groundwater that can be removed sustainably (*sensu* Bierkens

and Wada, 2019) and with the stable isotope information we currently have in hand, we are not able to meaningfully interpreting data for site-specific estimates, much less can we make regional estimates. This situation needs immediate attentions given that around 4,500,000 people in the YP rely on groundwater for drinking, irrigation, industrial, and urban uses (INEGI, 2019; CONAGUA, 2018). The availability of water in the YP has been estimated based on meteorological data within administrative boundaries, not at a landscape level or with isotopic, geological, hydrogeological, or geomorphological

considerations. This is an area of opportunity where isotopic data can help to fill this gap in information. In the YP, there is no surface drainage, and thus, no well-defined watersheds; yet, there are hydrogeological units which are (presumably) hydraulically connected, whose lateral and vertical limits are conventionally defined for purposes of evaluation, management and administration of national groundwater (Ley de Aguas Nacionales 2016). This definition is used when estimating the mean annual groundwater availability and it is not recognized as a management sub-unit in the hydrological-administrative

definitions. We consider that this type of inconsistency and disparity in criteria are causing issues in water management, particularly for water extraction and concessional volumes without clear, science-based hydrogeological criteria. This lack of information is largely caused by the inherent difficulty of delimitate groundwater divides, recharge and discharge areas. In some cases, large numbers of groundwater concessions are located in areas with more than 1200 mm of precipitation per year; however, there are areas with high density of concessions where precipitation is between 600 and 800 mm (Fig. 8). The

disparity among number of concessions and concessional volume (*i.e.* groundwater abstraction) together with reduced precipitation under current climatic change predictions (Lyra et al., 2017), might pose additional hydric stress onto areas with high population, agricultural activities, and touristic destinations such as Merida, Cancun, and the Riviera Maya. Should we have more data such as the amount of precipitation and the isotopic composition of groundwater, the isotopic composition of groundwater discharges and precipitation, we could make the calculations to help determine water allocations.



## 4 Future research

Water stable isotopes provide valuable information for assessing groundwater infiltration, recharge, instantaneous discharge, and residence time. The current lack of extensive water isotope data from the YP precludes us from fully understanding the regions groundwater situation. One future approach could be water sampling along a long N-S cross-section of the YP, including the highest altitudes in the Peninsula (~120 m in the Ticul range and ~220 m in the south) that might yield a trend similar to the one observed in Northern Central America (Lachniet and Patterson 2009), and might provide information about the altitudinal effect in areas with very low relief. The specifics of a sampling transect like this are difficult to define such as the appropriate number of sites and samples to achieve sufficient resolution. We consider it desirable to collect meteoric water on a grid pattern for at least two hydrological cycles. This approach would result in regional and local meteoric water lines needed for accurate estimates of local water availability, and be useful in hydrogeological and eco-hydrological studies.

Currently, we are developing a more detailed meteoric water line of the YP, using 11 locations distributed in the states of Campeche (n=4), Quintana Roo (n=4) and Yucatan (n=3). With those results, we will obtain several local meteoric water lines and the Regional (*Peninsular*) Meteoric Water Line; all of them assisting in improving water balance estimates of the aquifers in this Administrative Region. However, it is necessary to perform preliminary estimates of water availability on each of the aquifers and as hydrological sub-basins based on meteorological and isotopic data, evapotranspiration, natural discharge, concessional volumes and considering provision for environmental flow for each one of the identified hydrogeological units. This approach will assist in distinguishing regional water balances, differential water availability by regions and identify regions with current and future hydric stress. It is necessary to identify and protect recharge areas and areas where water extraction should be better regulated.

### Conclusions

We produced a regional meteoric water line for the Peninsula of Yucatan that better represents locations north of 20 ° N. We consider that the observed distribution of $\delta^{18}O$ and *d*-excess represents the seasonal variability and meteorological conditions in which meteoric water was collected; for instance, convective rains in the dry season, cold front events from November to March, and the passing of air masses with different moisture sources. Groundwater and its isotopic value has been more intensively studied and information published so far suggests that groundwater isotope composition is a combination of fast recharge after precipitation, water mixture due to surface interception and evaporation from lakes and sinkholes. Perennial and deciduous trees can use water from the vadose zone, from the aquifer and even from unconsolidated rock. The current stable water isotopes data provide information from specific sites, preventing interpretation, water planning and management at a regional scale.



**Code availability - N.A.**

**Data availability**

Data available at http://dx.doi.org/10.17632/wnz7my6y5r.1. Concessions and volumes available at
http://datos.aguaparatodos.org.mx/concesiones/.

**Appendices - N.A.**

**Team list - Author contribution**

EE, RMLB and HEM compiled the isotope data. GAG performed statistical and SIG analyses. EE prepared the manuscript
with contributions from all co-authors.

**Competing interests**

The authors declare that they have no conflict of interest.

**Disclaimer - N.A.**

**Acknowledgements**

CONACYT project CB286049 and Catedras CONACYT project 2944 (Water cycle modelling of the Peninsula of Yucatan).
Daniel Rios-Ponce helped with data analysis for concessional volumes. Eugene Perry provided important comments to the
manuscript. Mark Brenner and Jason Curtis granted access to databases and provided helpful suggestions. J. Curtis proofread
the manuscript.

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






**Figure 1: Locations in the Yucatan Peninsula with isotopic data ($\delta^2$H and/or $\delta^{18}$O) in meteoric water, groundwater, lakes and coastal lagoons. State boundaries and hydrological administrative regions are also showed.**






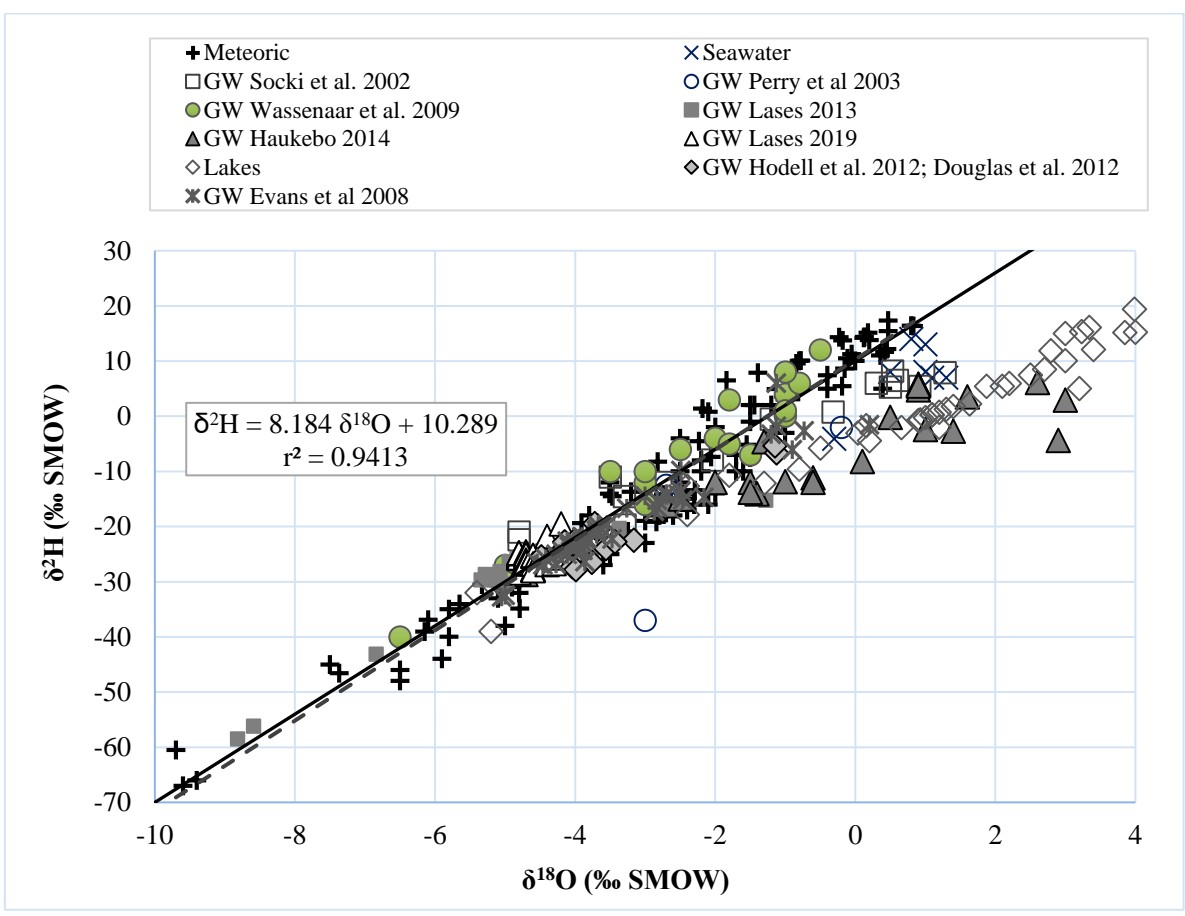

**Figure 2: Regional meteoric water line (dashed line) of the northern Peninsula of Yucatan. Seawater, lakes and groundwater isotopic composition in the YP. Data available in Supplementary Material 1.**





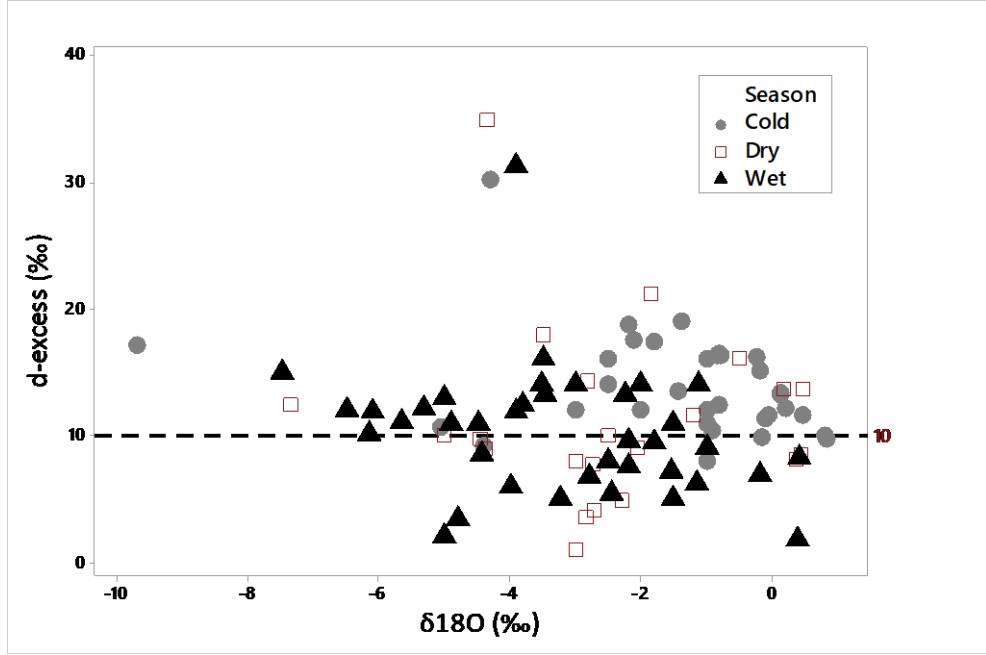

**Figure 3:** *d*-excess plot considering the three representative seasons in the Peninsula of Yucatan. Cold season from November to February, dry season from March to May, wet season from June to October.



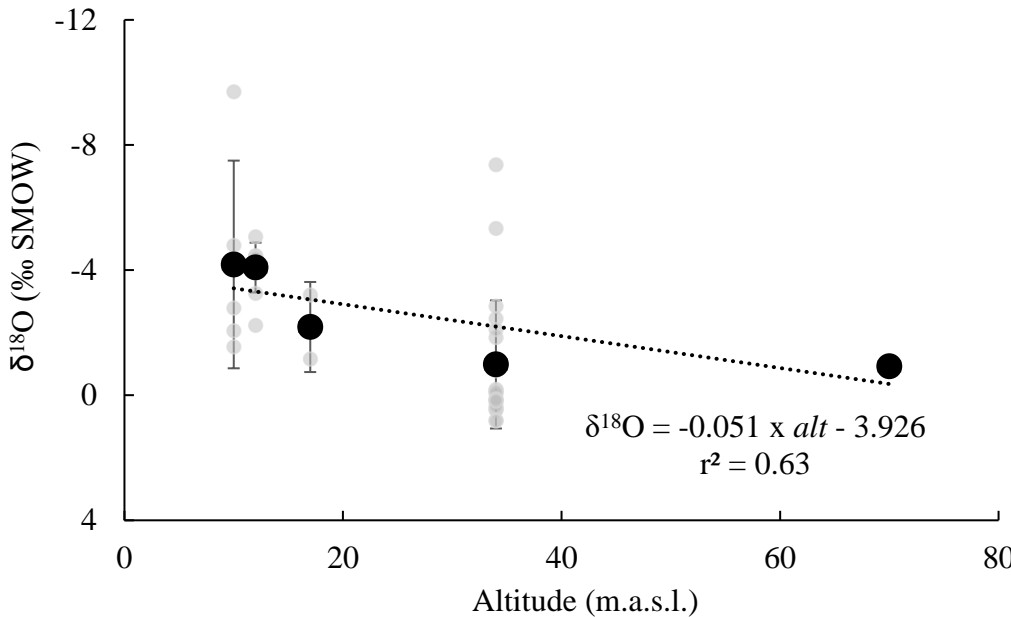


**Figure 4: Altitudinal effect in a north-south cross-section in the Peninsula of Yucatan. Altitude increase from 6 m.a.s.l (north) to ≈70**

**m.a.s.l (south).**





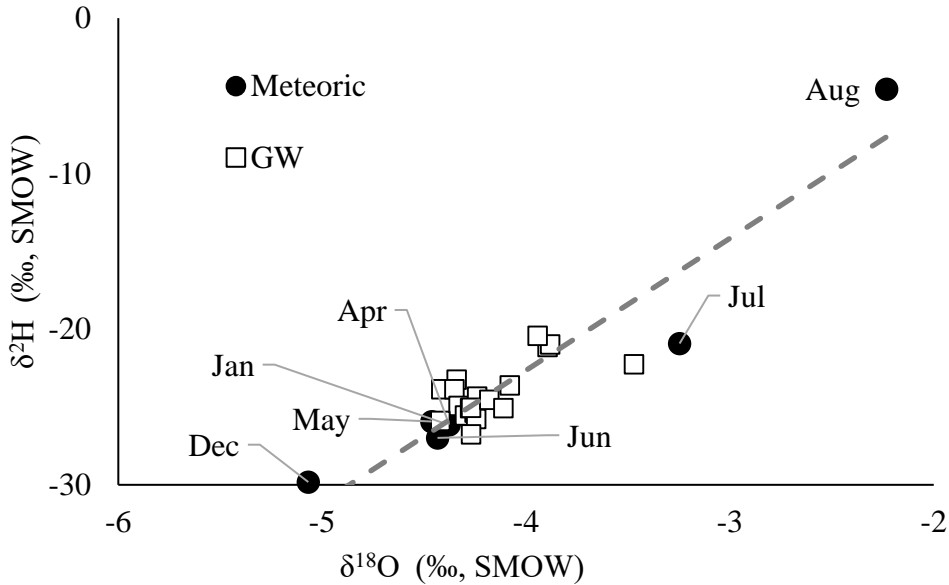

**Figure 5: Groundwater isotopic composition in meteoric water (circles) and groundwater (rectangles) in San José Tzal (NW of the Yucatan Peninsula).**





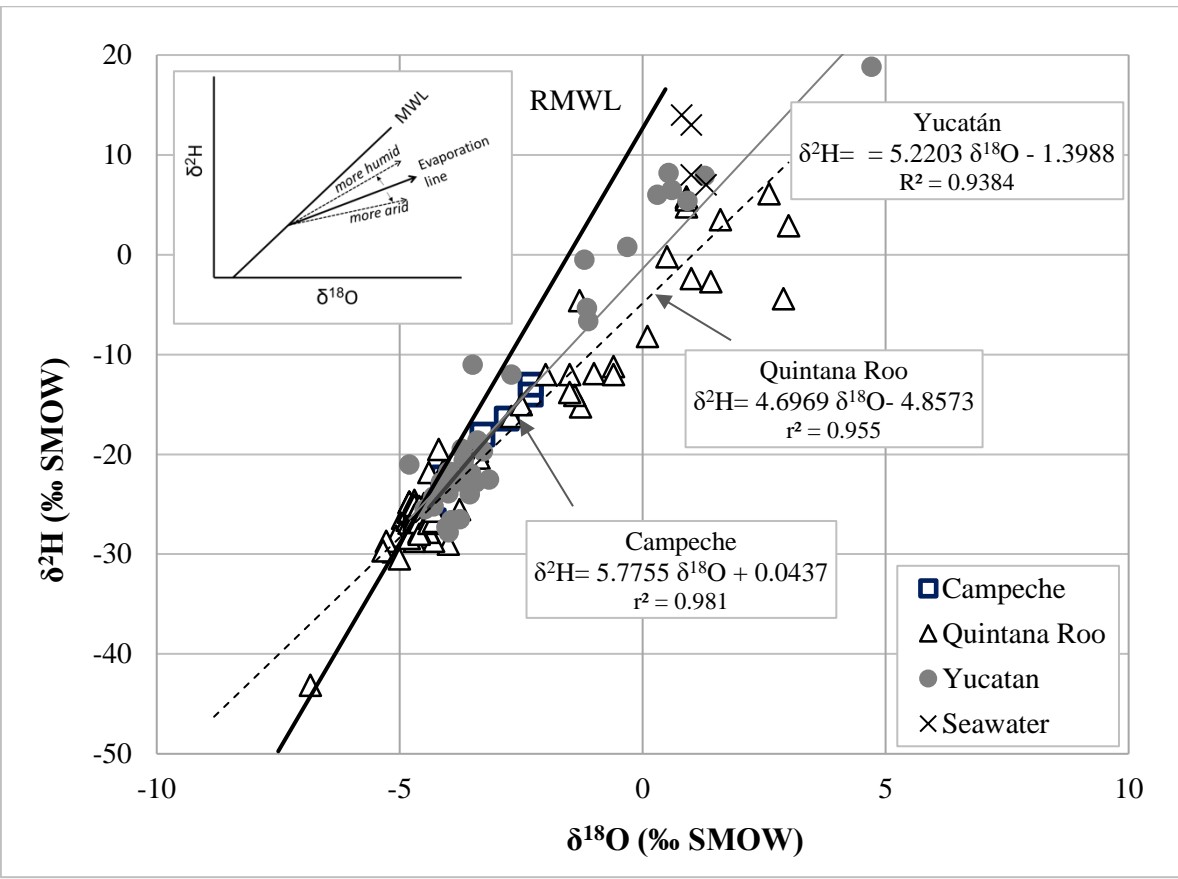

**Figure 6: Isotopic composition of groundwater relative to the Regional Meteoric Water line (RMWL) of the Yucatan Peninsula.**




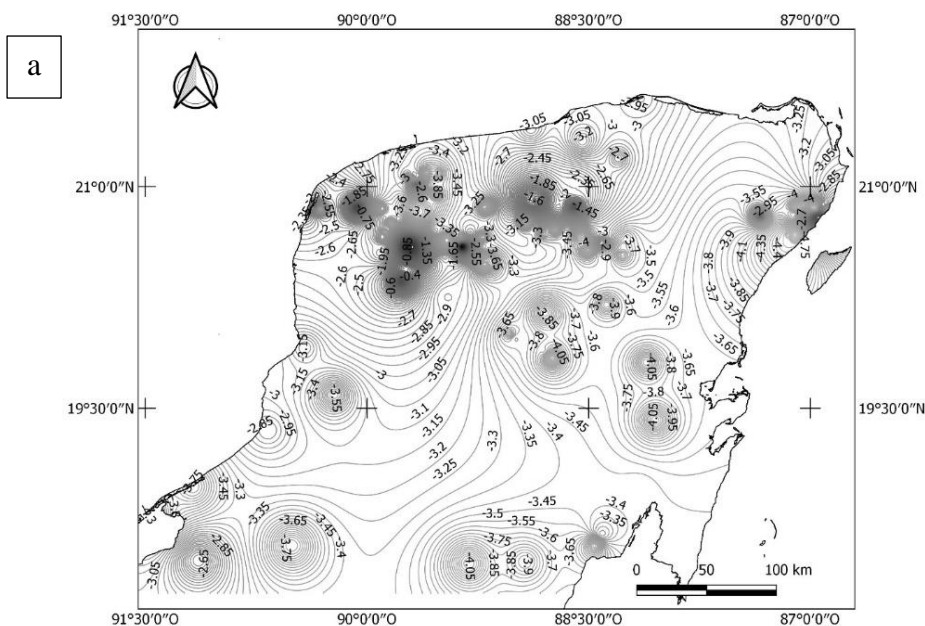

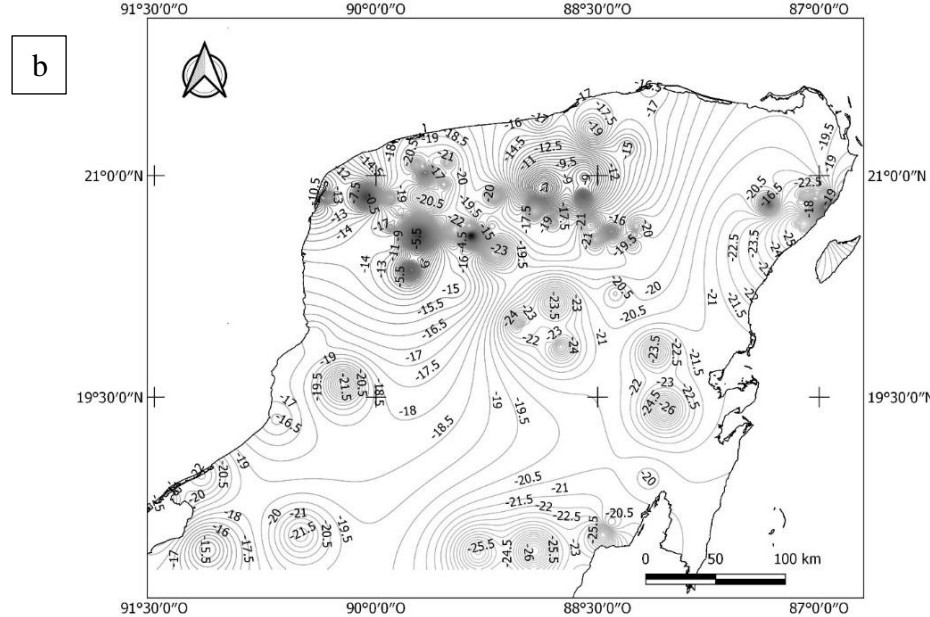

**Figure 7: Groundwater isoscape of the Yucatan Peninsula. a) δ¹⁸O and b) δ²H.**







**Figure 8: Map showing groundwater concessions distributed by sub-basin and by state (with number of concessions) relative to the historical annual precipitation (mm from 1910 to 2009)**