# Peer review of "Water stable isotopes ( $\delta^2$ H and $\delta^{18}$ O) in the Peninsula of Yucatan, Mexico."

_Hydrology and Earth System Sciences, 2020_

## Author Comment (AC1) · 1 Apr 2020

Lachniet and Patterson (2009) is the text but not in the reference list. Lachniet, M.S. and Patterson, W.P.: Oxygen isotope values of precipitation and surface waters in northern Central America (Belize and Guatemala) are dominated by temperature and amount effects. Earth and Planetary Science Letters, 284(3-4), 435-446, 2009.

---

## Referee Comment (RC1) · Anonymous Referee #1 · 24 Apr 2020

**General comment**

This manuscript focuses on presenting isotopic data of precipitation, groundwater and soil water sampled in the Peninsula of Yucatan, Mexico. The authors determined the regional meteoric water line for the study area, and discussed the origin of precipitation, as well as the variability in the isotopic signature of groundwater. I think the topic of this manuscript is potentially interesting for the readers of the journal, and I appreciated the effort done by the authors for the compilation of the isotopic dataset, which is based on a literature search and the collection of unpublished data. However, the paper needs a careful revision because the topic and the objectives of this study are not well presented in the introduction, the results are poorly described, and some discussion sections are disconnected from the rest of the manuscript. Furthermore, I think the

manuscript would benefit from a careful revision of English, due to the presence of some awkward terms.

Specific comments

- The current introduction section (lines 22-39) presents basic definitions and a generic explanation of the first applications of stable isotopes in tracer hydrology. I think the authors should narrow down the introduction to the specific topic of the manuscript, and clearly identify the research gaps (more recent references should be added as well). Furthermore, section 1.1 only highlights research gaps, specific of a study area, whereas the readers of Hydrology and Earth System Sciences could be more interested in a manuscript with a focus that is not only strictly-related to the specific study site.

- The main and the specific objectives of the manuscript are not clearly presented. I think the authors should reformulate lines 50-57 (page 2).

- I think the manuscript lacks a table summarizing the isotopic data compiled by the authors (e.g., water type, sample size, sampling location and period, and descriptive statistics should be reported in the manuscript).

- The results reported in the figures are poorly described in the text (particularly Fig. 4-7). Probably the manuscript would benefit from a separation of the results (I encourage to expand the description of the findings) and the discussions (these should be re-organized).

- Section 3.3 introduces for the first time that there are sap isotopic data sampled in the study area, but they are not reported in the manuscript. Since this discussion is quite disconnected from the previous and the following sections, I suggest to remove it from the manuscript.

- Section 3.5 (but it should be 3.4) also seems quite disconnected from the other sections in the manuscript, and most of it (particularly lines 199-212) is not very meaningful.

- Page 3, lines 73-75: It is unclear how the authors evaluated evaporation lines, and why they used evaporation lines for interpolation (spatial or temporal interpolation?) and comparison with the isotopic composition of groundwater.

- Page 5, lines 138-139: It is unclear how the authors could conclude that groundwater has a fast recharge by the examination of just few data reported in Fig. 5.

- Page 5, line 143: How did the authors determine that the groundwater follows the reported evaporation lines? I suggest to the authors to consider their results in light of recent findings reported in Benettin et al. (2018).

- Pages 5-6, lines 155-160: Based on the data and results reported in this manuscript, the inferred processes seem very speculative. I think the authors should remove these sentences or report the proper references supporting their statements.

- Figure 1: Where is located the state of Campeche? Please report the label in the map.

- Figure 7: Please increase the size of the labels.

- Figure 8: Symbols representing concessions are too small.

Technical corrections

- Page 1, line 15: Please replace "pore-waters" with "soil waters".

- Page 1, line 25: "$\delta$-plot" Probably the authors mean "a dual-isotope plot".

- Page 2, line 47: "is a representation".

- Page 2, line 55: Please explain the acronym "RMWL".

- Page 3, line 82: Please remove "Quantum GIS" and refer only to "QGIS 3.8".

- Page 3, line 84: It should be "Inverse Distance Weighted".

- Page 3, line 91: "studied less": Please revise this awkward sentence.

- Page 4, line 95: It is unclear what the authors mean with "that matrix".

- Page 5, line 135: It should be "analyses", however, the sentence is quite unclear.

- Page 8, line 228: Please replace "cycles" with "years".

References

Benettin P., Volkmann T.H.M., von Freyberg J., Frentress J., Penna D., Dawson T.E., Kirchner J.W., 2018. Effects of climatic seasonality on the isotopic composition of evaporating soil waters. Hydrology and Earth System Sciences, 22(5), 2881–2890. DOI: 10.5194/hess-22-2881-2018

---

## Author Comment (AC2) · 7 May 2020

General comment

The paper needs a careful revision because the topic and the objectives of this study are not well presented in the introduction, the results are poorly described, and some discussion sections are disconnected from the rest of the manuscript.

REPLY: We appreciate the comments and we have considered all of your suggestions. Overall, we agree with all of the comments from Referee #1 because our paper presents information that contributes to the knowledge of the study of water isotopes, beyond its application in the region. Our most important result is the meteoric water line. We cannot be conclusive with the data in hand but we can hypothesize as far as the data and the existing literature allow us to do. We restrict ourselves to the statements supported by our data.

The Anonymous Referee # 1 suggested us that it should include a clear objective and we have rewritten the section so the Objective is well presented (Page 2, lines 51-53)

Furthermore, I think the manuscript would benefit from a careful revision of English, due to the presence of some awkward terms.

REPLY: We have completed edition for proper English language, grammar, punctuation, spelling, and overall style by one or more qualified native English-speaking editors.

Specific comments

- The current introduction section (lines 22-39) presents basic definitions and a generic explanation of the first applications of stable isotopes in tracer hydrology. I think the authors should narrow down the introduction to the specific topic of the manuscript, and clearly identify the research gaps (more recent references should be added as well). Furthermore, section 1.1 only highlights research gaps, specific of a study area, whereas the readers of Hydrology and Earth System Sciences could be more interested in a manuscript with a focus that is not only strictly-related to the specific study site. REPLY: We appreciate the suggestion and we rewrote the Introduction section with less general information about isotopes, narrowing down to what we consider essential information and recent references. In order to be of ample interest for readers of HESS and other in the world, we have deleted the section 1.1, so our contribution is not very parochial. We stress that we aim to "contribute with to the state of the art of meteoric water isotopes that will assist in ecohydrological, paleoclimatic, physiological and other research, not only in Mexico, but also in the Great Caribbean Area and other locations with similar latitudinal and geological conditions" (Page 2, lines 51-53)

- The main and the specific objectives of the manuscript are not clearly presented.

[Figure]

I think the authors should reformulate lines 50-57 (page 2). REPLY: Completed and rewritten with the above mentioned corrections and suggestions from the Reviewer (Page 2, lines 51-57).

- I think the manuscript lacks a table summarizing the isotopic data compiled by the authors (e.g., water type, sample size, sampling location and period, and descriptive statistics should be reported in the manuscript). REPLY: Thank you for the keen observation. Table 1 has been added and it summarizes the main characteristics of the isotopic data compiled: n, States, Locations, Sources (References), Range of dates, Water types, Aquifers, Hydrogeological sub-regions, Altitude (m.a.s.l.) and $\delta$18O- $\delta$ 2H Provider).

- The results reported in the figures are poorly described in the text (particularly Fig. 4-7). Probably the manuscript would benefit from a separation of the results (I encourage to expand the description of the findings) and the discussions (these should be re-organized). REPLY: We appreciate and consider the comments. We have included better description of the findings of the results presented in the figures, but we prefer to maintain the format as Results and Discussion together as they are now, because we can present the data and immediately discuss and present our arguments and interpretation with the data on hand. The specific comments for some section are answered lines below, responding each specific comment.

- Section 3.3 introduces for the first time that there are sap isotopic data sampled in the study area, but they are not reported in the manuscript. Since this discussion is quite disconnected from the previous and the following sections, I suggest to remove it from the manuscript. REPLY: We apologize for the mistake in the sub-heading number. We understand the suggestion of removing the section as the reviewer considers it "disconnected from the previous and the following sections". We prefer to maintain the section because not only contributes to support the objective (presents most of the existing data of stable water isotopes), but also allow us to present an interpretation of what could be happening in regards of water used by plants, namely the use of soil water as other published literature have found. In order to attend the observation "they are not reported in the manuscript", we have mentions sap isotopic data in the introduction (Page 2, lines 46-49). Discussion (Section 3.4, Page 7, lines 194-212) contrast and compare the finding in the YP with other reports regarding water transpired by plants in karstic aquifers and other ecosystems. We kindly ask to consider our arguments and discussion, properly connecting all section of the manuscript regarding the integration of water isotopic data for studies of water balances and ecohydrology.

- Section 3.5 (but it should be 3.4) also seems quite disconnected from the other sections in the manuscript, and most of it (particularly lines 199-212) is not very meaningful. REPLY: We understand the comment and mostly agree with it. We have restructured the complete section as a section of the discussion named "3.5 Stable isotopes for better groundwater management" before future research, as the use of water stable isotopes helps to fill gaps in information regarding sustainable groundwater in karstic areas with large extraction volumes. Thanks for the observation and we expect the Reviewer finds satisfactory the modification (Page 7 Lines 214-223).

- Page 3, lines 73-75: It is unclear how the authors evaluated evaporation lines, and why they used evaporation lines for interpolation (spatial or temporal interpolation?) and comparison with the isotopic composition of groundwater. REPLY: We apologize for the misunderstanding; we present groundwater evaporation lines that include data with both $\delta 18O$ and $\delta 2H$ data, and in cases where only one number was provided (for instance $\delta 18O$ in Perry papers) we provide a "theoretical" deuterium value by interpolating the expected $\delta 2H$ using the existing evaporation line. We have reworded Section 2.1 and provided a reference of the use of linear interpolation so it is clear what we present in the Supplementary material 2 (Lines 71-76). In addition, the results and Discussion section mentioning the evaporation lines (Page 5, Lines 145-162) is extended to provide deeper interpretation of Figure 6.

- Page 5, lines 138-139: It is unclear how the authors could conclude that groundwater has a fast recharge by the examination of just few data reported in Fig. 5. REPLY: The

Reviewer is right on the observation that, with so little data, we cannot be conclusive. We did not express this sentence as a proved fact, we are merely hypothesizing of what this data set suggest and would be required to verify. We have rewritten the paragraph so it is now an observation based on the data we currently have and compared to published research with similar outcomes (Page 5, Lines 139-143)

- Page 5, line 143: How did the authors determine that the groundwater follows the reported evaporation lines? I suggest to the authors to consider their results in light of recent findings reported in Benettin et al. (2018). REPLY: We thank the Reviewer for providing such valuable reference. We have rewritten the paragraph presenting our point of view and support it based on the results published by Benettin and discuss our limited dataset under the light of those findings (Page 5, Lines 148-162)

- Pages 5-6, lines 155-160: Based on the data and results reported in this manuscript, the inferred processes seem very speculative. I think the authors should remove these sentences or report the proper references supporting their statements. REPLY: We agree with the Reviewer and we removed those sentences since we have provided our argument in Page 5, Lines 136-143. Thanks for the observation.

- Figure 1: Where is located the state of Campeche? Please report the label in the map. Change completed. State labels included and slight changes in legend nomenclature.

- Figure 7: Please increase the size of the labels. Change completed

- Figure 8: Symbols representing concessions are too small. REPLY: Considering the comments and focusing on the objective, the authors have agreed on removing figure 8 and all the associated text through the manuscript.

Technical corrections:

- Page 1, line 15: Please replace "pore-waters" with "soil waters". Change completed

- Page 1, line 25: "$\delta$-plot" Probably the authors mean "a dual-isotope plot". Change completed

- Page 2, line 47: "is a representation". Change completed

- Page 2, line 55: Please explain the acronym "RMWL". Change completed

- Page 3, line 82: Please remove "Quantum GIS" and refer only to "QGIS 3.8". Change completed

- Page 3, line 84: It should be "Inverse Distance Weighted". Change completed

- Page 3, line 91: "studied less": Please revise this awkward sentence. REPLY: Change completed. The sentence is now "whereas meteoric water (27%) and lakes (27%) have been less sampled and analysed." (Page 3, line 88-89)

- Page 4, line 95: It is unclear what the authors mean with "that matrix". REPLY: We wanted to describe the environment or compartment in which the water samples were collected. For example, soil water or groundwater. We have rewritten the sentence "sampling efforts have been focused on that compartment of the hydrosphere" referring to aquifers. (Page 4, line 95). Thanks.

- Page 5, line 135: It should be "analyses"; however, the sentence is quite unclear. REPLY: Thank for noticing this. We wanted to say, "Groundwater has been slightly more intensively sampled and most reports represent single samples analysed for specific purposes". We have corrected the sentence (line 137).

- Page 8, line 228: Please replace "cycles" with "years". Change completed

References Benettin P., Volkmann T.H.M., von Freyberg J., Frentress J., Penna D., Dawson T.E., Kirchner J.W., 2018. Effects of climatic seasonality on the isotopic composition of evaporating soil waters. Hydrology and Earth System Sciences, 22(5), 2881–2890. DOI: 10.5194/hess-22-2881-2018

Table 1. Summary of the main characteristics of the isotopic data compiled in the Peninsula of Yucatan, Mexico.

| Category | Descriptor |
|---|---|
| States | 3[a] |
| Data points (locations) | 170[a] |
| Data sources (references) | 17 |
| Range of dates | Aug 1973 - Aug 2018 |
| Aquifers | 3[a] |
| Administrative regions | 4[a] |
| Altitude (m.a.s.l.) | 0 – 210 |
| Isotopic composition of water (‰) [b] | |
| Meteoric $\delta^{18}$O (n=128) | -2.74 (-9.7, 0.83) |
| Meteoric $\delta^2$H (n=124) | -11.36 (-67, 17.33) |
| Groundwater $\delta^{18}$O (n=213) | -3.19 (-8.82, 6.81) |
| Groundwater $\delta^2$H (n=199) | -18.66 (-58.52, 28.17) |
| Seawater $\delta^{18}$O (n=7) | 0.77 (-0.3, 1.3) |
| Seawater $\delta^2$H (n=7) | 7.56 (-4.1, 14.0) |
| Coastal lagoon $\delta^{18}$O (n=2) | 2.1 (1, 3.2) |
| Coastal lagoon $\delta^2$H (n=2) | 5.39 (-0.29, 11.07) |
| Lake $\delta^{18}$O (n=126) | 1.8 (-5.4, 5.82) |
| Lake $\delta^2$H (n=110) [c] | 4.69 (-39, 23.21) |
| Mangrove $\delta^{18}$O (n=1) | 6.1 |
| Mangrove $\delta^2$H (n=1) | 29.2 |
| $\delta^{18}$O - $\delta^2$H analyses provider | 11[d] |

[a] See in Figure 1

[b] Isotopic data are in per mil (‰) mean, minimum and maximum.

[c] Missing $\delta^2$H data interpolated from evaporation lines. See Supplementary Material 2.

[d] ARES Division, Johnson Space Center; Atomic Energy Comission Contract AT (30-1)3204; Auburn, Alabama (ANIMAL); Department of Geological Sciences, University of Florida; Geoscience Mass-Amherst; Insituto Mexicano de Tecnologia del Agua; National Geophysical Data Center, Boulder - Colorado; Stable Isotope Facility of the University of California at Davis; Stable Isotope Laboratory Environment Canada; Texas A&M Galveston; Yale University.

Fig. 1.

[Figure]

[Figure]

Figure 7: Groundwater isoscape of the Peninsula of Yucatan. a) δ¹⁸O and b) δ²H.

**Fig. 2.**

---

## Referee Comment (RC2) · Anonymous Referee #2 · 19 May 2020

This technical note presents a dataset of stable isotopes in precipitation, groundwater, and lake water in the Yucatan Peninsula (YP), Mexico compiled from published and unpublished information. Using the compiled data the authors defined a regional meteoric water line for the YP and attempt to evaluate the spatial variability of the stable isotopic composition of groundwater in the region. Even though I appreciate the authors' effort, the lack of information regarding the depth of collection of the groundwater samples hampers the usefulness of the dataset and does not allow for an accurate inference of groundwater recharge and flow processes. Because of this important constraint, I consider the dataset will be of little to no use in future studies. In addition, the manuscript presents several major methodological and data interpretation issues that limit its suitability to be considered for publication in HESS. I describe these issues in detail below.

[Figure]

Major issues:

An analysis of the factors influencing the stable isotopic composition of precipitation was not carried out. I am not sure how the authors came to the conclusions in P1 L16-19. Supporting such statement would require a thorough investigation of the potential sources and trajectories of moisture contributing to local/regional precipitation (e.g., using Lagrangian back trajectory models) and the estimation of the fractions of convective (and stratiform) precipitation among other types of analyses.

The search literature protocol should be described in more detail. In general, the Boolean operators (AND, OR) and other type of operators (e.g., between and within) must be used to connect keywords (e.g., water, isotope, hydrology, etc.) appropriately. A thorough literature review should clearly indicate the used operators and how they were used to connect the search terms. In addition, since the study region is located in a Spanish speaking country, it is likely that valuable information has been published in scientific journals published in that language. Thus, the search should also be done using terms in Spanish and include literature databases that publish literature in that language such as Scielo, Latindex, Redalyc for completeness.

There is an important limitation with the compiled groundwater data. That is, in most of the cases the depth at which those data were obtained is not reported. I checked the Supplementary Material 1 and only 18% out of the 213 groundwater data reported this information and the depth of collection varied largely between 1 and 120 m for the cases in which this was included. The former issue hampers the correct interpretation of the groundwater isotopic data, while the latter indicates that constructing a groundwater isoscape that does not take into account the sampling depth is not meaningful. Unfortunately, these limitations hamper the utility of the compiled dataset for better understanding the groundwater hydrology of the YP in this and future studies.

As stated by the editor in the first revision of the manuscript, the basics of isotope hydrology are overemphasized and the value of the compiled data and related inferences

are not clear for the reader. For example, the basics on the factors that influence the temporal variability of the isotopic composition in precipitation is substantially described in lines P4 L108-117, but only a couple of sentences are regarded to the interpretation of the compiled data (P4, L117-119; in addition the reference Li et al. 2015 is missing from the list of references).

In their response letter to the previous revision by the editor the authors claim that the major finding of their work is the determined RMWL. However, the presented RMWL resembles well one previously reported for the study region (P4 L102-103). Thus, I do not find anything novel about this finding.

The authors mention a supposed altitude effect. Such effect is only apparent (i.e., the stable isotopic composition of rainfall decreasing as altitude increases) because the y-axis of the figure is in reverse order than in the other isotopic plots presented in the manuscript. Even though it is not clear what the data presented in Fig. 4 represent (this must be clearly described in the caption), it is very complicated to argue for a potential altitude effect given the very small range (< 100 m) in elevation among the sampling stations. In addition, I consider that the information presented is not relevant at all for the evaluation of the altitude effect since at some elevations there is a lot of information (6 and 27 m elevation) in comparison to others (a single value at 70 m elevation). I thus suggest the authors to reconsider the statements in P5 L129-133 accordingly.

The "retention of the meteoric of isotopic signature in groundwater that suggests fast recharge after precipitation" is argued. Although the aforementioned interpretation of the data could feasible depending on the hydrogeological characteristics of the groundwater reservoir, I am not sure how the data presented in Fig.5 alone supports this argument. This is because it is not clear when the groundwater samples were collected in comparison to the precipitation ones, and because the characteristics of the groundwater reservoir are not described. A through interpretation of the data would require the consideration of hydrogeological characteristics of the aquifer.

Even though it is clear that groundwater recharge is influenced by evaporation effects, the geographical location of the monitoring sites for itself (north versus south) cannot explain the differences among the stable isotopic composition of water in the different states of the YP (P5 L141-142). The variability depends on the specific features of each of the monitoring locations (e.g., type and spatial distribution of the soil/s, hydrogeological features, presence of open water bodies, etc.). However, since such information is not presented, it is not possible for the authors to explain the observed evaporative effects in the isotopic signal of groundwater across the study region. Because of this, most of their inferences in P5-P6 L144-170 are strongly speculative and are unsubstantiated by the presented data.

In open water bodies the lighter water isotopes are preferentially removed during evaporation, so the remaining liquid phase is enriched in the heavier stables isotopes of water (O18 and D). Thus, the inference of the authors in P6 L173-174 is erroneous.

The isotopic data presented in the figures is not sufficiently described and discussed (e.g., the GW, lakes, and seawater data in Fig. 1; the d18O versus d-excess variability across seasons in Fig. 2; the variability of the stable isotopic data at different elevations in Fig. 3).

Sections 3.3, 3.5 (section 3.4 is missing) and 4 are not directly related to the compiled dataset. Section 3.3 presents a summary of literature findings related to vegetation water uptake and sections 3.5 and 4 read rather like commentaries.

Minor issues:

P1 L22-24: Delete as these concepts are widely known by isotope hydrologists.

P1 L35: What do you mean by "input parameters"?

P1 L55: Pore water data is not presented.

P3 L63: Describe what is the "Virtual library and Catalogue" and why it was included for the literature search.

P3 L67: Specify that most of the compiled groundwater data did not include information about the sampling depth.

P3 L74: Specify what proportion of the data was interpolated to estimate the d2H values.

P3 L83-86: As mentioned above, I consider that constructing an isoscape without accounting for the depth of collection of the samples does not provide relevant information. I thus suggest to delete this analysis from the manuscript.

P4 L95: What does "on that matrix" refers to?

Figure 1: The map is confusing. It is not clear whether the text within the map refers to the states or the administrative regions. As the states are often referred to in the manuscript, I would suggest that the states names are shown in the figure instead of the administrative regions. Or even better, to present both with the text in different colors for the states and the administrative regions matching the line colors presented in the legend of the figure (red and black)

Figure 2: Specify what the circles of different colors and sizes represent as well as the error bars. This presents data from different water sources published in different studies but those are never discussed in the manuscript. I suggest to describe and discuss all the data presented in the figure or simply not showing it.

Figure 4: Describe what the dots of different color and size as well as the error bars represent.

Technical corrections:

P2 L31: change "d coefficient" by "d-excess" and use the latter consistently in the whole manuscript.

Correct the number of the subsections in section 3 (i.e., there are 2 subsections 3.2).

Figure 2: specify what the solid black line in the plot represents.

Figure 5: Delete the word "Groundwater" placed before "isotopic composition...".
Specify what the dashed line represents.

---

## Author Comment (AC3) · 4 Jun 2020

Anonymous Referee #2 Received and published: 19 May 2020

This technical note presents a dataset of stable isotopes in precipitation, groundwater, and lake water in the Yucatan Peninsula (YP), Mexico compiled from published and unpublished information. Using the compiled data the authors defined a regional meteoric water line for the YP and attempt to evaluate the spatial variability of the stable isotopic composition of groundwater in the region. Even though I appreciate the authors' effort, the lack of information regarding the depth of collection of the groundwater samples hampers the usefulness of the dataset and does not allow for an accurate inference of groundwater recharge and flow processes.

REPLY: thanks to the Referee #2 for all the comments. We would like to begin by saying that this effort represent (to our knowledge) the first attempt to put together most (if not all) of the isotopic data in water in this region, it will be a baseline of this tropical karstic area with scarce data on stable isotopes for water management. The limitation we face would be the same for any researcher who would attempt to make sense of the little data available; yet, the effort has to be done as a starting point in order to build a research program such as the one in the make by the authors for the years to come. This Technical Note is put together for any researcher who may take this database in the future as reference, and then move forward towards a better, deeper analysis using stable water isotopes. Second, we tried not to be conclusive in any way regarding the data and interpretation that we present. The data that we present from the PY is likely all of the available data and we did our best interpreting the number. However, we agree that, due to the limitations of the database, our interpretation might change or it may be confirmed once more data is available. We clearly state where the information gaps are and what is required to fill the void. We agree that we cannot make accurate inference of groundwater recharge and flow processes; thus we did not try to go further. We think that clearly noticing and acknowledging the lack of information is one of the most important motors for research in emergent countries. May this paper be an invitation to the scientific community to research more intensively this important tropical karstic area of the world.

Major issues:

An analysis of the factors influencing the stable isotopic composition of precipitation was not carried out. I am not sure how the authors came to the conclusions in P1L16-19. Supporting such statement would require a thorough investigation of the potential sources and trajectories of moisture contributing to local/regional precipitation (e.g., using Lagrangian back trajectory models) and the estimation of the fractions of convective (and stratiform) precipitation among other types of analyses.

REPLY: We consider that the sentence is not a conclusion as the Referee men-
tions; rather, we draw the general panorama of the most possible factors influencing the stable isotopic composition of precipitation, based on evidence from regions with similar climatic conditions. The study published by Hu and Dominguez in 2015 (DOI: 10.1175/JHM-D-14-0073.1) tracking sources of moisture in North America, mentioned that both oceanic and terrestrial moisture (i.e. evapotranspiration) are important sources of precipitation. Their work addresses the North American Monsoon; however, we understand that moisture from the Atlantic Ocean have a westward movement, likely depositing precipitation in the YP. We understand and agree with the Referee that "potential sources and trajectories of moisture contributing to local/regional precipitation (e.g., using Lagrangian back trajectory models) and the estimation of the fractions of convective (and stratiform) precipitation among other types of analyses", would provide depth and certainty to our suggestions. However, we think that such analyses are more on the realm of the atmospheric sciences and are out of the scope of this Technical note. Nonetheless, this keen observation opens up a panorama and an opportunity for us that we have not reflected before. We appreciate the suggestion and we would consider a future contribution dealing with sources and trajectories of moisture and precipitation in the Yucatán Peninsula. Finally, as the referee notes, we do not have enough data for modeling and it will be out of the scope of this Technical note.

The search literature protocol should be described in more detail. In general, the Boolean operators (AND, OR) and other type of operators (e.g., between and within) must be used to connect keywords (e.g., water, isotope, hydrology, etc.) appropriately. A thorough literature review should clearly indicate the used operators and how they were used to connect the search terms. In addition, since the study region is located in a Spanish speaking country, it is likely that valuable information has been published in scientific journals published in that language. Thus, the search should also be done using terms in Spanish and include literature databases that publish literature in that language such as Scielo, Latindex, Redalyc for completeness.

REPLY: Thanks for the suggestion. We have completed literature search in Sci-
elo and Redalyc as suggested. Latindex is a platform that provides qualitative information about academic journals, print and online; thus, was not useful directly as search engine. Unfortunately, we did not found matches for its addition. Thus, Datamining was conducted using search engines Google Scholar, Scopus, Internet Archive, Web of Science, Scielo, Redalyc and the "Virtual library and Catalogue" at www.cicy.mx/biblioteca/biblioteca-virtual using the search terms Isotopes, Water, Yucatan, Campeche and Quintana Roo and their equivalents in Spanish Isótopos, Agua. Boolean operators (AND, OR) were used to connect keywords appropriately. A total of 17 publications were identified with data of the required characteristics and quality (P3 L62-67)

There is an important limitation with the compiled groundwater data. That is, in most of the cases the depth at which those data were obtained is not reported. I checked the Supplementary Material 1 and only 18% out of the 213 groundwater data reported this information and the depth of collection varied largely between 1 and 120 m for the cases in which this was included. The former issue hampers the correct interpretation of the groundwater isotopic data, while the latter indicates that constructing a groundwater isoscape that does not take into account the sampling depth is not meaningful. Unfortunately, these limitations hamper the utility of the compiled dataset for better understanding the groundwater hydrology of the YP in this and future studies.

REPLY: We understand the concern and we realize the complexity of the situation. It is true that we present an isoscape in which few data has a depth of water sampling associated, and those number came from one paper were deep sampling was completed (Socki et al. 2002). The isoscape previously published by Wassenaar et al (2007) reported groundwater sampled form shallow and unconfined wells, presumably between 5 and 20 m in depth. They do not have certainty of the real depth of all samples since sometimes they collected water from household taps, cisterns, storage tanks and even springs. The expectation was that "all of the water wells sampled were screened in phreatic aquifers". The same assumption applies here; we produced an isoscape with
water sampled at phreatic level. We understand the concern that disparity on data "hampers the correct interpretation of the groundwater isotopic data"; so, we amended the isoscape not using the data deeper than 30 m below ground level published by Socki et al. (2002), so the isoscape hereby presented represents mostly phreatic level or depth to screened wells (frequently around 20 m). It is not the ideal scenario for interpreting data, but it is what we can do with what we have on hand, available, and with adequate scientific quality (P3 L81-84 and current Figure 6).

As stated by the editor in the first revision of the manuscript, the basics of isotope hydrology are overemphasized and the value of the compiled data and related inferences are not clear for the reader.

REPLY: As suggested by both referees, we rewrote the Introduction section with less general information about isotopes, narrowing down to what we consider essential information and recent references.

For example, the basics on the factors that influence the temporal variability of the isotopic composition in precipitation is substantially described in lines P4 L108-117, but only a couple of sentences are regarded to the interpretation of the compiled data (P4, L117-119; in addition the reference Li et al. 2015 is missing from the list of references).

REPLY: We consider that our interpretation is not reduced to a couple of sentences, all the paragraph (P4 L112-124) refers to the compiled data. We interpret the data in the PY in the light of other published research, contrasting and comparing them. We present the argumentation in this manner because we outlined the possible scenarios occurring in the area and then stressing those situations most likely occurring based on the reduced data we have, going as far into educated interpretation as possible. We appreciate Referee #2 for noting the missing references. It is now added. Thanks much.

In their response letter to the previous revision by the editor the authors claim that the major finding of their work is the determined RMWL. However, the presented RMWL

HESSD
resembles well one previously reported for the study region (P4 L102-103). Thus, I do not find anything novel about this finding.

REPLY: We do not think redundant to provide a RMWL that resembles a local meteoric line representing one location, Playa del Carmen (Quintana Roo, Mexico, 20°35.2', -87°8.04'). Our approach to this apparent dull comparison is that, in this particular case, a local meteoric water line might represent a greater geographical area; yet, having more data makes the RMWL better suited to be the meteoric line. We are not sure if the non-different meteoric lines in this case are just coincidence or are common elsewhere, but we think that when our database contain more information, these meteoric lines will be different. We do not completely agree with the lack of novelty of this finding, it is rather intriguing. We are currently preparing a paper containing an original database that represents the meteoric water line of the PY, including 11 locations distributed in the three states. We can only mention that it has a different slope (lower than 8.1) and an intercept higher than 11.6. Therefore, as abovementioned, this technical note is the starting point of a research program in the make.

The authors mention a supposed altitude effect. Such effect is only apparent (i.e., the stable isotopic composition of rainfall decreasing as altitude increases) because the y-axis of the figure is in reverse order than in the other isotopic plots presented in the manuscript. Even though it is not clear what the data presented in Fig. 4 represent (this must be clearly described in the caption), it is very complicated to argue for a potential altitude effect given the very small range (

now as follows "When exploring the altitude effect in a north-to-south cross section, we cannot observe nor suggest a trend of the  $\delta$ 18O at elevations higher than 100 m.a.s.l. because the current isotopic data represents only an area with low relief. Perhaps sampling the longest N-S cross-section, including the highest altitudes in the Peninsula ( $\approx$  120 m in the Ticul range and  $\approx$ 220 m in the south), might yield a trend as the one observed in Northern Central America (Lachniet and Patterson 2009)". (P5 L134-138). In addition, Figure 4 has been eliminated, as it does not contribute substantially to the discussion.

The "retention of the meteoric of isotopic signature in groundwater that suggests fast recharge after precipitation" is argued. Although the aforementioned interpretation of the data could feasible depending on the hydrogeological characteristics of the groundwater reservoir, I am not sure how the data presented in Fig.5 alone supports this argument. This is because it is not clear when the groundwater samples were collected in comparison to the precipitation ones, and because the characteristics of the groundwater reservoir are not described. A through interpretation of the data would require the consideration of hydrogeological characteristics of the aquifer.

REPLY: Both referees coincide in the observation that, with so little data, we cannot be conclusive. We did not express this sentence as a proven fact, we are merely hypothesizing of what this data set suggest and would be required to verify. Certainly, both precipitation and rainfall data were collected simultaneously from May 2007 to April 2008. In this case, we drilled and artesian well, specifically to collect groundwater samples. We have rewritten the paragraph so it is now an observation based on the data we currently have and compared to published research with similar outcomes (P5 L140-154). Even though it is clear that groundwater recharge is influenced by evaporation effects, the geographical location of the monitoring sites for itself (north versus south) cannot explain the differences among the stable isotopic composition of water in the different states of the YP (P5 L141-142). The variability depends on the specific features of each of the monitoring locations (e.g., type and spatial distribution of the
soil/s, hydrogeological features, presence of open water bodies, etc.). However, since such information is not presented, it is not possible for the authors to explain the observed evaporative effects in the isotopic signal of groundwater across the study region. Because of this, most of their inferences in P5-P6 L144-170 are strongly speculative and are unsubstantiated by the presented data. REPLY: We deeply appreciate referee #2 for providing such an important insight. We now include information regarding soils, geomorphologic features and open water bodies, and relate them to what we consider most likely evaporative effects in the isotopic signal as represented by the proposed evaporation lines (P5 L156- P6 L171).

In open water bodies the lighter water isotopes are preferentially removed during evaporation, so the remaining liquid phase is enriched in the heavier stables isotopes of water (O18 and D). Thus, the inference of the authors in P6 L173-174 is erroneous. REPLY: We appreciate the observation, we had a typo here, and we meant "less depleted values". It is now corrected (P7 L209).

The isotopic data presented in the figures is not sufficiently described and discussed (e.g., the GW, lakes, and seawater data in Fig. 1; the d18O versus d-excess variability across seasons in Fig. 2; the variability of the stable isotopic data at different elevations in Fig. 3).

REPLY: Thanks for the observation, Referee #1 expressed the same concern. We have included better description in the text of the findings and the results presented in the figures. We assume that referee #2 mentions figures 2 to 4. The GW, lakes, and seawater data is not discussed directly in the RMWL section, for it is only provided there for comparison purposes, and a specific section of groundwater is presented in section 3.2. Regarding the d18O versus d-excess variability across seasons, we restrict ourselves to the statements supported by our data (P4 L116-121). We present some educated guesses from L121 to L124 such as convective rains occur more during the dry season, cold front events common from November to March and the wide distribution of d-excess in wet season (July to October) once the reader has seen the

HESSD
figure. Finally, we have eliminated former Figure 4 (altitudinal effect) as per suggestion of referee #2 "I consider that the information presented is not relevant at all for the evaluation of the altitude effect since at some elevations there is a lot of information (6 and 27 m elevation) in comparison to others (a single value at 70 m elevation)"

Sections 3.3, 3.5 (section 3.4 is missing) and 4 are not directly related to the compiled dataset. Section 3.3 presents a summary of literature findings related to vegetation water uptake and sections 3.5 and 4 read rather like commentaries.

REPLY: We apologize for the mistake in the sub-heading number. In order to attend the observations, we have improved the Introduction and Discussion so we can contrast and compare the finding in the PY with other reports regarding water transpired by plants in karstic aquifers and other ecosystems. We have properly connected all section of the manuscript regarding the integration of water isotopic data for studies of water balances and ecohydrology. Section 3.5 has been re-structured named "Stable isotopes for better groundwater management" before future research (4), addressing how the use of water stable isotopes helps to fill gaps in information regarding sustainable groundwater in karstic areas with large extraction volumes. Thanks for the observation and we expect referee #2 finds satisfactory the modification (Sections 3.4 and 3.5).

Minor issues:

P1 L22-24: Delete as these concepts are widely known by isotope hydrologists. Change completed.

P1 L35: What do you mean by "input parameters"? According to Gat (2005) input parameters is the precipitation, the basic supply for meteoric lines. We have now added precipitation for clarity purposes (P2 L30).

P1 L55: Pore water data is not presented. After Referee #1 suggestion, we change the term for soil water (P2 L54). Soil water is later interpreted as the water used by plants
for transpiration.

P3 L63: Describe what is the "Virtual library and Catalogue" and why it was included for the literature search. This is the search engine in the library of the Yucatan Center for Scientific Research (CICY). We include it because it has access to thesis and dissertation published at all of the Government research centers and major Universities in Mexico.

P3 L67: Specify that most of the compiled groundwater data did not include information about the sampling depth. Completed P3 L71.

P3 L74: Specify what proportion of the data was interpolated to estimate the d2H values. Completed. It represents 7% of the  $\delta$ 2H data, P3 L78.

P3 L83-86: As mentioned above, I consider that constructing an isoscape without accounting for the depth of collection of the samples does not provide relevant information. I thus suggest to delete this analysis from the manuscript. We have corrected the analysis, including only shallow groundwater (not considering deep samples P3 L82-84). As we mentioned above, previous efforts neither have certainty of the real depth of all samples and the main assumption is that isoscapes are produced with water sampled at phreatic level.

P4 L95: What does "on that matrix" refers to? We wanted to describe the environment or compartment in which the water samples were collected. For example, soil water or groundwater. We have rewritten the sentence "sampling efforts have been focused on that compartment of the hydrosphere" referring to aquifers (P4 L97). Thanks.

Figure 1: The map is confusing. It is not clear whether the text within the map refers to the states or the administrative regions. As the states are often referred to in the manuscript, I would suggest that the states names are shown in the figure instead of the administrative regions. Or even better, to present both with the text in different colors for the states and the administrative regions matching the line colors presented
in the legend of the figure (red and black) Change completed. State labels included and changes in legend nomenclature.

Figure 2: Specify what the circles of different colors and sizes represent as well as the error bars. This presents data from different water sources published in different studies but those are never discussed in the manuscript. I suggest to describe and discuss all the data presented in the figure or simply not showing it. We think referee #2 is addressing Figure 4. Figure 2 has the legend for all circles and does not have error bars. As it pertains to Figure 4, we agree with the comment as answered above and Figure 4 has been eliminated, as it does not contribute substantially to the discussion.

Figure 4: Describe what the dots of different color and size as well as the error bars represent. Figure 4 has been eliminated.

Technical corrections:

P2 L31: change "d coefficient" by "d-excess" and use the latter consistently in the whole manuscript. Thanks for the correction, referee #1 also mentioned it and the change has been completed (P1 L26), d-excess is used consistently the whole manuscript.

Correct the number of the subsections in section 3 (i.e., there are 2 subsections 3.2). Completed

Figure 2: specify what the solid black line in the plot represents. Solid black lines represents the GMWL, added to the Figure caption.

Figure 5: Delete the word "Groundwater" placed before "isotopic composition. . .". Specify what the dashed line represents. Changes completed, thanks for the observation. Figure capture has been updated.

Please also note the supplement to this comment: https://www.hydrol-earth-syst-sci-discuss.net/hess-2020-16/hess-2020-16-AC3supplement.pdf Interactive comment

**HESSD**